# Impact of the Vaginal and Endometrial Microbiome Pattern on Assisted Reproduction Outcomes

**DOI:** 10.3390/jcm10184063

**Published:** 2021-09-08

**Authors:** María del Carmen Diaz-Martínez, Andrea Bernabeu, Belén Lledó, Concepción Carratalá-Munuera, Jose A. Quesada, Francisca M. Lozano, Vicente Ruiz, Ruth Morales, Joaquín Llácer, Jorge Ten, Juan Carlos Castillo, Adoración Rodríguez, Rauf Nouni-García, Adriana López-Pineda, Belén Moliner, Rafael Bernabeu

**Affiliations:** 1Instituto Bernabeu Reproductive Medicine, 03016 Alicante, Spain; maykadiaz84@gmail.com (M.d.C.D.-M.); abernabeu@institutobernabeu.com (A.B.); vruizcamara@institutobernabeu.com (V.R.); jllacer@institutobernabeu.com (J.L.); jten@institutobernabeu.com (J.T.); jcastillo@institutobernabeu.com (J.C.C.); drodriguez@institutobernabeu.com (A.R.); bmoliner@institutobernabeu.com (B.M.); rbernabeu@institutobernabeu.com (R.B.); 2Molecular Laboratory, Instituto Bernabeu Biotech, 03016 Alicante, Spain; blledo@institutobernabeu.com (B.L.); plozano@institutobernabeu.com (F.M.L.); rmorales@institutobernabeu.com (R.M.); 3Clinical Medicine Department, School of Medicine, University Miguel Hernández de Elche, 03550 Alicante, Spain; jquesada@umh.es (J.A.Q.); raufrng@gmail.com (R.N.-G.); adriannalp@hotmail.com (A.L.-P.)

**Keywords:** microbiome, pregnancy rates, repeated implantation failure, reproductive medicine

## Abstract

Uterine microbiota may be involved in reproductive health and disease. This study aims to describe and compare the vaginal and endometrial microbiome patterns between women who became pregnant and women who did not after in vitro fertilization. We also compared the vaginal and endometrial microbiome patterns between women with and without a history of repeated implantation failures (RIF). This pilot prospective cohort study included 48 women presenting to the fertility clinic for IVF from May 2017 to May 2019. Women who achieved clinical pregnancy presented a greater relative abundance of *Lactobacillus* spp. in their vaginal samples than those who did not (97.69% versus 94.63%; *p* = 0.027. The alpha and beta diversity of vaginal and endometrial samples were not statistically different between pregnant and non-pregnant women. The Faith alpha diversity index in vaginal samples was lower in women with RIF than those without RIF (*p* = 0.027). The alpha diversity of the endometrial microbiome was significantly higher in women without RIF (*p* = 0.021). There were no significant differences in the vaginal and endometrial microbiomes between pregnant and non-pregnant women. The relative abundance of the genera in women with RIF was different from those without RIF. Statistically significant differences in the endometrial microbiome were found between women with and without RIF.

## 1. Introduction

Technological advances in mass sequencing have enabled the identification of different microbial communities in the uterine cavity, including in the vagina and the endometrial cavity [1]. A recent review [2] concluded that there was not enough evidence for a “core” or bacterial resident population in the uterus, and the existence of uterine microbiota might be reflective of bacterial tourists or invaders. Nevertheless, uterine microbiota may be involved in reproductive health and disease [1,2]

The main bacteria at the vaginal and endometrial level belong to the genus *Lactobacillus*—producers of lactic acid that maintain the acidic pH of the vagina, which acts as a barrier against pathogens [3]. The association between vaginal flora and pregnancy outcomes has been widely studied for years. The live birth rate is correlated with the production of H_2_O_2_ by *Lactobacillus* spp., and inversely correlated with the existence of bacterial vaginosis. Thus, alterations to the vaginal flora—for example, due to bacterial vaginosis (provoked by *Gardnerella vaginalis*)—are associated with an increased risk of miscarriage [4,5]. Other pathogenic microorganisms—such as *Chlamydia trachomatis*, *Neisseria gonorrhoeae*, and *Mycoplasma tuberculosis*—may cause subclinical alterations related to risk factors for subfertility [6].

The prevalence of infertility is on the rise [7], and assisted reproductive technologies (ART) are increasingly in demand, as well as being safer and more successful [8]. Endometrial implantation is the single most important event determining the success of embryo transfer in ART [9]. Other factors include the microbial colonization of the upper genital tract [10]—and possibly the uterine–cervical canal—which has been shown to be a significant, independent determinant of the success of assisted reproductive treatments [11]. Recently, the reproductive tract microbiota have been associated with embryo transfer failure [12].

Until now, the reproductive microbiome has been studied by analyzing samples taken from either the endometrium or the vagina. Although the uterine microbiota appear to be a continuum from the vaginal microbiota, previous studies have found differences between the endometrial and vaginal microbiota [13]. Currently, the role of these microbiota in embryo implantation and pregnancy outcomes is unclear. One of the main obstacles to determining the bacterial composition of the endometrium is that the small amount of the initial sample makes it vulnerable to contamination with exogenous bacterial DNA.

Recently, Di Simone et al. [14] suggested that classification based on endometrial bacterial patterns could help prevent obstetric complications through personalized treatments. In light of scientific evidence that alterations to the reproductive microbiome reduce women’s fertility by negatively impacting on embryo implantation, determining the bacterial composition of the vagina and the endometrium may contribute to improving the prognosis of fertility treatments. Thus, the primary objective of the present study was to describe and compare the vaginal and endometrial microbiome patterns between women who became pregnant and women who did not after in vitro fertilization (IVF). The secondary objective was to compare the vaginal and endometrial microbiome patterns between women with and without a history of repeated implantation failures (RIF).

## 2. Materials and Methods

### 2.1. Design and Study Population

This pilot prospective cohort study took place in a private assisted reproduction clinic. It included women presenting to the fertility clinic for IVF from May 2017 to May 2019 who met the following inclusion criteria: aged 18–50 years; undergoing frozen embryo transfer (FET) with euploid embryos, using either their own or donated gametes; and an indication to use the intracytoplasmic sperm injection (ICSI) method to generate embryos. Exclusion criteria were: the use of antibiotics in the three months preceding the fertility treatment, uterine malformations, untreated hydrosalpinx, known implantation failure factors, or unwillingness to sign informed consent.

Participants followed the usual ART procedure with ovarian stimulation and ICSI per protocol, and underwent preimplantation genetic testing for aneuploidies (PGT-A) at the blastocyst stage using NGS (VeriSeq, Illumina,, San Diego, California, USA) with the MiSeq Sequencer (Illumina). One euploid embryo was transferred in the cycle following the ovarian stimulation via ultrasound-guided transfer, in accordance with established protocols. Endometrial preparation via estrogen–progesterone replacement therapy was performed. Human chorionic gonadotropin (β-hCG) blood testing was performed 8–9 days after the embryo transfer to test for pregnancy.

The local ethics committee approved this study in January 2017 (Reference code: 16/318), and it was conducted in accordance with the Declaration of Helsinki. All participants signed written informed consent to take part in the study.

### 2.2. Study Variables

The following baseline characteristics were collected on enrollment: age (years), weight (kg), tobacco use (yes/no), number of previous pregnancies, previous miscarriages (yes/no), number of miscarriages, previous fertility treatments (yes/no), RIF, i.e., failed implantation after ≥ 3 transfer cycles with good-quality embryos (yes/no), donated oocytes (yes/no), donated semen (yes/no), and normozoospermia diagnosis after semen analysis (yes/no). The date and the endometrial thickness (mm) were recorded before the embryo transfer.

The primary study outcome was clinical pregnancy (yes/no), i.e., pregnancy evidenced by ultrasound of fetal cardiac activity after positive β-hCG. Other reproductive outcomes included the result of the β-hCG blood test (positive/negative), biochemical miscarriage (yes/no), and clinical miscarriage (yes/no).

### 2.3. Sample Collection

Vaginal samples were collected at different stages of IVF treatment: (1) during the secretory phase of the cycle previous to FET (days 18–22 of the cycle), (2) on the day of the embryo transfer, and (3) on the day of the pregnancy test. A dry swab was used to collect vaginal fluid from the bottom of the posterior sac by direct visualization using the vaginal speculum, with the patient in the lithotomy position. In order to avoid contamination, we did not use lubricant or gel on the speculum. On the day of the embryo transfer, we took the sample before preparing the embryo transfer to prevent interference by the procedure.

Endometrial samples were collected during the secretory phase of the cycle previous to FET (days 18–22 of the cycle), and we used the Tao Brush IUMC Endometrial Sampler; this device minimizes the risk of contamination during the collection of the endometrial sample via a sheath that closes prior to withdrawal from the uterus. All vaginal and endometrial samples were stored at −80 °C until further analysis.

### 2.4. Sample Analysis

Microbiome patterns were analyzed by estimating the prevalence and variability of types of bacteria at both the vaginal and endometrial levels. We used metagenomics for sample analysis, studying the 16S rRNA gene marker of the included samples via next-generation sequencing (NGS). Analyses took place in the molecular genetics laboratory of the fertility clinic. Rigorous controls were carried out for the reagents and all of the equipment used in all of the steps carried out during the processing and analysis of the samples.

### 2.5. DNA Extraction

DNA extraction was performed using the PureLink Microbiome DNA Purification Kit (Thermo Fisher, PureLink^TM^ Microbiome DNA Purification Kit, Darmstadt, Germany and/or its affiliates). The DNA was quantified using a Qubit 2.0 Fluorometer (Thermo Fisher). The extracted DNA was stored at −20 °C for later use.

### 2.6. Amplification of Region V3V4 of the 16S rRNA Gene

Polymerase chain reaction (PCR) amplification of the variable region V3V4 of the 16S rRNA gene was performed using Taq DNA polymerase (2× KAPA HiFi HotStart, Roche, Rotkreuz, Switzerland) in the presence of dNTPs, as well as oligonucleotides 357F and 806R, at a final concentration of 1 μM and an average of 100 ng of DNA, and at a final reaction volume of 25 μL, following the recommendations of Illumina (16S Metagenomic Sequencing Library Preparation). PCR was carried out in a thermal cycler (Verity, Applied Biosystems, San Francisco, CA, USA). For the validation of the PCR technique, all amplification reactions included positive and negative controls without DNA templates. The PCR products were visualized using agarose electrophoresis, verifying that the amplified DNA band was the correct size (449 base pairs). All products of amplification were stored at −20 °C for subsequent sequencing.

### 2.7. Sequencing of Region V3V4 of 16S rRNA Gene

Once the V3V4 amplicon was obtained and purified, we generated the library with the identifying indices of each sample using the Nextera XT sequencing kit (Illumina). After the purification of the libraries, the samples, which were previously diluted to a concentration of 4 nM before being mixed and prepared for sequencing, were quantified using a Qubit 2.0 Fluorometer (Thermo Fisher). The final concentration of the library was 15 pM. The library was sequenced using MiSeq Reagent Kit v3 (Illumina) reagents. We used MiSeq (Illumina) as the sequencing equipment and for the metagenomics of the workflow.

### 2.8. Bioinformatic Analysis of the Sequences

The primary analysis of the obtained sequences consisted of demultiplexing, using the MiSeq Reporter software (Illumina). The unindexed paired-end sequences of each sample were exported from the MiSeq system in FASTQ format for the analysis. The bioinformatic analysis of the sequences was carried out using the QIIME2 package. Further data analysis was also performed with MicrobiomeAnalyst software. Deblur was used to filter and denoise the sequences with QIIME2. The sequences were grouped in operational taxonomic units (OTUs) with a similarity percentage of 97%.

In order to estimate alpha diversity—i.e., the number of different species present in the sample—a rarefaction analysis was performed on 1000 sequences per sample, followed by an alpha diversity analysis. To aid in comparisons of alpha diversity, differences in library sizes across samples were adjusted by the rarefaction method. Three different indices were used: Shannon, Simpson, and Faith. Since these indices did not follow a normal distribution, the non-parametric Mann–Whitney U method was used. The Shannon index quantifies the different types of taxa present in the community, and considers the richness and equitability of species. If two sites are equally rich in species, however, with one site dominated by a single species and the other showing more evenness, the second would clearly be considered more diverse. The Simpson index expresses the probability that two microorganisms randomly selected from an infinitely large community are of different species.

The results for beta diversity were visualized with QIIME2 using the graphics generated by principal coordinates analysis (PCoA), obtained with EMPeror. We carried out the analysis of beta diversity using the unweighted UniFrac index. Beta diversity expresses composition in terms of the abundance of different taxa among the samples. UniFrac is a measure of beta diversity that uses phylogenetic information to compare samples belonging to the interest groups—in this case, four; the unweighted version is qualitative. Therefore, UniFrac measures concordance based on the abundance of OTUs in each sample, including phylogenetic distances. The matrices with beta diversity measurements were analyzed using PERMANOVA for differences in composition according to the group they belonged to (type of sample).

Taxonomic assignment was performed using a classification based on a filtering of the 99_otus sequence from the Greengenes database to the V3V4 region. Finally, we performed the univariate analysis of each specified taxon or group according to the results we obtained, using the correction for multiple testing.

### 2.9. Statistical Analysis

A descriptive analysis of all variables was performed by calculating frequencies for the qualitative parameters and minima, maxima, means, and standard deviations (SD) for the quantitative variables. To compare them, we used the parametric Student’s *t*-test or the non-parametric Mann–Whitney U test, as appropriate. Microbiome patterns as well as vaginal and endometrial samples were compared between groups according to outcome variables, using double-entry tables for qualitative variables and the chi-squared test. For quantitative variables, Student’s *t*-test was used to assess the association between microbiome patterns and clinical pregnancy outcomes. A linear mixed-effects model was constructed to determine the evolution of the vaginal microbiome pattern and its association with clinical pregnancy rate and RIF diagnosis. Analyses were performed using R v.3.5.1.

## 3. Results

The study included 48 participants, who provided 192 samples, from which 189 sequences suitable for analysis were obtained. Figure 1 details the number of women and samples initially included in the study, along with losses and net inclusion data and analysis. Table 1 shows the baseline characteristics of the study participants, whose mean age was 39.44 years. A total of 26 women (54.2%) tested positive on the β-hCG pregnancy test during the study, and 21 achieved clinical pregnancy (43.8%); of these, 38.9% (*n* = 8) had a history of miscarriages, compared to 70.0% (*n* = 19) of those who did not get pregnant (*p* = 0.034). The biochemical and clinical miscarriage rates were 10.4% (*n* = 5) and 14.28% (*n* = 7), respectively.

### 3.1. Differences between Vaginal and Endometrial Microbiome Patterns

We found statistically significant differences in the alpha diversity of endometrial versus vaginal microbiomes (*p* = 0.014 for the Shannon index and *p* = 0.046 for the Simpson index), with higher values in endometrial samples (Figure 2a). Using PERMANOVA, the beta diversity of the samples showed statistical differences in composition between vaginal and endometrial microbiomes (*p* = 0.001). The unweighted UniFrac PCoA revealed a clear pattern of separation between vaginal and endometrial samples (Appendix A). The endometrial samples are grouped at the extreme right of the graph. The percentage of the variance explained by each component is shown on the axes (principal component (PC)1: 34.16%; PC2: 20.31%; PC3: 8.58%). The first and second components could together explain more than 50% of the variability between the samples.

Regarding the taxonomic characterization of the samples during the secretory phase of the cycle previous to FET, there was a clear dominance of the genus *Lactobacillus* in both the vaginal and the endometrial microbiomes. The relative frequencies of the most abundant genera grouped by sample type (vaginal/endometrial) are shown in Figure 2b. Microbiome profiles showed relative differences in genera and species present in the vaginal and endometrial samples. The univariate analysis reached statistical significance for *Lactobacillus* spp., *Streptococcus* spp., *Ureaplasma* spp., *Delftia* spp., *Anaerobacillus* spp., and *L. helveticus*. Several genera were more abundant in the vagina than in the endometrium: *Lactobacillus*, *Streptococcus,* and *Ureaplasma*. The remaining genera were more abundant in the endometrium (Table 2). Appendix A shows the relative frequency of the most abundant species for each sample type. *Lactobacillus iners* presents a higher relative abundance in endometrial samples, without significant difference (64% versus 40% in vaginal samples). There was a significant difference in the abundance of *L. helveticus*: 28% in the endometrium versus 47% in the vagina (*p* = 0.013).

### 3.2. Evolution of the Vaginal Microbiome

There were no statistically significant differences in alpha or beta diversity between the samples over the different timepoints. In regard to taxonomic characterization, no statistically significant differences between timepoints for the composition of either genera or species were found (Figure 3). There were some apparent changes in the abundance of the genera *Lactobacillus*, *Streptococcus*, and *Prevotella*: both *Lactobacillus* and *Streptococcus* were more abundant in vaginal samples collected during the secretory phase of the cycle and on the day of the embryo transfer, showing a decrease on the day of the pregnancy test; on the other hand, *Prevotella* showed a high abundance during the secretory phase of the cycle, and even higher on the day of the pregnancy test. However, the univariate analysis showed no statistically significant differences. At the species level, we found some differences in relative abundance for the following species: *L. helveticus*, *L. iners*, *L. gasseri*, and *L. jensenii* (Figure 3). The most abundant species were *L. helveticus* on the day of the embryo transfer, *L. iners*, during the secretory phase of the cycle, and *L. gasseri* on the day of the pregnancy test. On the day of the pregnancy test, results showed a smaller proportion of *L. jensenii*. However, none of these differences were statistically significant.

### 3.3. Microbiome Patterns Associated with Clinical Pregnancy after FET

#### 3.3.1. Vaginal Microbiome Pattern

Analyzing diversity as a function of the clinical pregnancy rate after FET, we found a greater alpha diversity in women who did not get pregnant, although the trend did not reach statistical significance (Shannon *p* = 0.075 and Simpson *p* = 0.086). Regarding the beta diversity of vaginal samples collected during the secretory phase of the cycle, we did not find statistically significant differences between women who became pregnant and those who did not. For the samples collected on the day of embryo transfer, we likewise found no statistically significant differences in alpha or beta diversity between these groups. However, there was a trend suggestive of a negative correlation between the clinical pregnancy rate and alpha diversity (*p* = 0.152). The vaginal samples taken on the day of the pregnancy test showed no differences in alpha or beta diversity.

Regarding taxonomic characterization, women who became pregnant presented a significantly greater abundance of *Lactobacillus spp*. in vaginal samples collected during the secretory phase of the cycle previous to FET, compared to those who did not get pregnant. On the other hand, *Streptococcus* spp. and *Prevotella* spp. were more abundant in the latter group (Appendix A). The differences were observed at the genus level for *Lactobacillus* spp. (91% with no gestation vs. 99% with gestation; *p* = 0.045) and at the species level for *L. reuteri* (0.39% vs. 0.17%; *p* = 0.040; Figure 4a).

Similar results were obtained for samples collected on the day of embryo transfer. Women who achieved clinical pregnancy presented a greater relative abundance of *Lactobacillus* spp. than those who did not (97.69% versus 94.63%; *p* = 0.027; Figure 4b and Appendix A); the opposite was true for the case of *Streptococcus* spp. (Appendix A). For vaginal samples collected on the day of the pregnancy test, the findings were similar (Appendix A). The univariate analysis showed statistically significant differences (*p* = 0.049) for the genus *Lactobacillus* spp. (99.74% with gestation versus 97.73% without) and the species *L. reuteri* (0.30% versus 0.15%, respectively; *p* = 0.059; Figure 4c).

#### 3.3.2. Endometrial Microbiome Pattern

Concerning the alpha and beta diversity of endometrial samples collected during the secretory phase of the cycle previous to FET, no statistically significant differences were found between pregnant and non-pregnant women.

Figure 5 shows the taxonomic characterization of endometrial samples according to pregnancy outcome; women who achieved a clinical pregnancy had a greater abundance of *Lactobacillus* spp., *Gardnerella* spp., *Burkholderia* spp., and *Anaerobacillus* spp.; in contrast, *Streptococcus* spp., *Ralstonia* spp., *Prevotella* spp., and *Delftia* spp. were more abundant in women who did not get pregnant; however, the univariate analysis did not show significant differences in the relative abundance of these genera between groups.

### 3.4. Microbiome Patterns Associated with Diagnosis of RIF

#### 3.4.1. Vaginal Microbiome Pattern

Regarding the vaginal microbiome pattern in the samples collected during the secretory phase of the cycle, we found no differences in alpha diversity between women with and without history of RIF according to either the Shannon or Simpson alpha diversity indices (Figure 6a). The results were statistically significant only for the Faith index. The box diagram for the Faith phylogenic alpha diversity index (phylogenetic analog of taxon richness expressed as the number of tree units found in a sample) yielded a *p*-value of 0.027, representing a significantly lower Faith alpha diversity index in women with RIF compared to women without RIF.

In relation to beta diversity, no statistically significant differences were observed between women with and without RIF diagnosis (Appendix A). Likewise, the univariate analysis showed no statistically significant results. In relation to the taxonomic allocation, women with an RIF diagnosis had a lower relative abundance of the genus *Streptococcus*, and a higher abundance of *Prevotella* spp., *Ureaplasma* spp., and *Dialister* spp. Women without RIF presented a higher relative abundance of *Streptococcus* spp., *Veillonella* spp., and *Aerococcus* spp. As for the genus *Lactobacillus*, no differences were observed between groups (Appendix A). At the species level, we found a higher relative abundance of *L. helveticus* in women with RIF, and of *L. iners*, *L. jensenii*, *L. gasseri*, and *L. agalactiae* in women without RIF (Appendix A).

#### 3.4.2. Endometrial Microbiome Pattern

The alpha diversity of the endometrial microbiome during the secretory phase of the cycle was significantly higher in women without RIF (Figure 6b; *p* = 0.021 for both Shannon and Simpson indices). There were also statistically significant differences in beta diversity, as shown in the PCoA graph (Appendix A). There is a clear pattern of separation between women with and without RIF: the samples collected from women diagnosed with RIF fall in the top center of the graph, while those collected from women without RIF are clustered in the center. The axes show the percentage of the variance explained by each component (PC1: 31%; PC2: 14.1%; PC3: 8.9%). The results for the first and second components explain more than 45% of the variability between study samples.

The taxonomic assignment in the frequency table (Appendix A) represents the relative abundance of the different taxa present in the samples according to RIF diagnosis. A greater abundance of the genus *Prevotella* was observed in women with RIF. In the univariate analysis, we found statistically significant differences for the genus *Ralstonia*, observing a much higher relative abundance in women without RIF compared to women with RIF (0.73% versus 0.09%; *p* = 0.001). Appendix A shows the differences in relative abundance at the species level. *L. iners* and *L. jensenii* were more abundant in women without RIF, while *L. helveticus* and *Sneathia amnii* had a larger presence in women with RIF.

### 3.5. Linear Mixed-Effects Model Tests by Pregnancy Outcome and RIF Diagnosis

The variation in alpha diversity (Shannon index) at different timepoints of the cycle was assessed (Figure 7), and there were no statistically significant differences in the evolution of the vaginal microbiome patterns between timepoints of the treatment or according to the achievement of clinical pregnancy (*p* = 0.412).

Analyzing this evolution by RIF diagnosis (*p* = 0.019), we observed a variation in the vaginal microbiome pattern over time in women without RIF. Specifically, these women showed a decrease in alpha diversity from the follicular to the luteal phase. In contrast, the women with RIF showed a stable microbiome pattern across different timepoints. This lack of dynamism in the pattern of the vaginal microbiome in women with RIF could entail a lack of adaptation to endometrial physiology and preparation and, therefore, a worse prognosis for embryo implantation (Figure 7).

## 4. Discussion

The results of this study show that there were no differences in the diversity of the vaginal microbiome between women who got pregnant and those who did not, but there were differences in its taxonomic characterization, and while statistically significant differences were not observed in the endometrial microbiome patterns, there were statistical differences in the composition of the vaginal versus endometrial microbiomes. Our results also suggest stability of the vaginal microbiome throughout different timepoints of the cycle. On the other hand, the vaginal samples collected during the secretory phase of the cycle previous to FET showed differences only for the Faith index between the women with and without RIF diagnosis, with a higher alpha diversity in women with a history of RIF. At the taxonomic level, the relative abundance of the genera in women with RIF was different from those without RIF. With regard to endometrial microbiome patterns, statistically significant differences were found between women with and without RIF.

Franasiak et al. [15] showed that the microbiome at the time of embryo transfer could be successfully characterized without altering standard clinical practice. However, our vaginal samples were taken with a dry swab, and endometrial samples by Tao Brush, in order to discern the microbiome from both locations and avoid possible contamination. With reference to the taxonomic characterization in our study, similarly to Franasiak et al. [15], we also found that the relative abundance of bacteria of the genus *Lactobacillus* was higher in vaginal samples of women who achieved clinical pregnancy after FET than of those who did not. However, Franasiak et al. [15] did not obtain statistically significant values as we did. We also observed that *Streptococcus* and *Prevotella* may be associated with a poor prognosis with regard to gestation, while an abundance of *Lactobacillus* spp. could be indicative of more favorable conditions. Since no statistically significant differences were obtained when analyzing the differences in relative abundance between samples taken at three different timepoints of the cycle, we concluded that the vaginal sample could be taken at any timepoint to obtain similar findings. The microbiome is usually affected by ovarian steroids, and the FET cycle may be related to higher local serum progesterone (P4) levels than in the previous cycle (timepoint 1). However, since all patients had one endometrial preparation per hormonal cycle substituted, we considered that this factor was homogeneous for all study participants.

*Lactobacillus* spp. are the most abundant bacteria in the vaginal samples. The ability of lactobacilli to inhibit infection without inducing inflammation can maximize fertility and favor pregnancy outcomes [16]. In relation to the endometrium microbiome, we did not find differences between women who got pregnant and those who did not. The presence of a non-*Lactobacillus*-dominated microbiome (NLDM)—i.e., < 90% *Lactobacillus* spp. with > 10% of other bacteria—was associated with lower rates of implantation, pregnancy rate, pregnancy progression, and live birth [17]. In this study, women classified as having an NLDM and showing a relative abundance of more than 80% *Lactobacillus* spp. in the endometrium showed good pregnancy outcomes, suggesting that this threshold could be considered sufficient for embryonic implantation [17]. In addition, even if classified as having an NLDM, the endometrium with a dominant quantity of *Bifidobacteria* could also be an acceptable environment for implantation [18]. Kyono et al. [19] analyzed both the endometrial and vaginal microbiomes in the infertile Japanese population and assessed their impact on embryonic implantation. The bacterial status of the endometrium and vagina showed *Lactobacillus*-dominated microbiota (> 90% *Lactobacillus* spp.) in the endometria of 38% of IVF patients, and in the vaginas of 44.3%. The mean percentage of *Lactobacillus* spp. in pregnant women was 96.45% (SD 33.61%). A considerably high proportion of NLDMs was found in the endometria of infertile Japanese women. A subsequent pilot study by Kyono et al. [18] did show that the predominance of *Lactobacillus* in the endometrium was favorable in terms of the pregnancy rate, but the results were not conclusive. A previous meta-analysis [20] concluded that women with abnormal vaginal microbiota are approximately 1.4 times less likely to achieve first-trimester pregnancy after IVF treatment compared to women with normal microbiota; however, the methodologies of the included studies were heterogeneous.

In the present study, the taxonomic characterization showed differences in the microbial profiles between the vaginal and endometrial samples in terms of genera and species. The univariate analysis performed on the relative abundance of the different genera by sample type reached statistical significance for *Lactobacillus* spp., *Streptococcus* spp., *Ureaplasma* spp., *Delftia* spp., *Anaerobacillus* spp., *Ralstonia* spp., and the species *L. helveticus*. These results suggest that vaginal and endometrial samples are different in terms of both microbial diversity and the composition of the taxa. In a previous pilot study [21], the dominant members of the microbial community were constant in the vagina and the cervix, and generally consistent in the endometrium; however, the relative proportions varied. In addition, infertile women tended to have *Ureaplasma* spp. in the vagina and *Gardnerella* spp. in the cervix more frequently than fertile women. In relation to the above-mentioned study, our analysis of alpha and beta indices for the microbial diversity of vaginal and endometrial samples revealed significant differences. Although the diversity of the vaginal microbiome could be expected a priori to be higher than that of the endometrial one due to factors such as more direct exposure to the outside of the body and direct contact with the seminal sample during sexual intercourse, the present study found a more diverse microbiome in the endometrium than in the vagina. This fact could be due to the non-dominance of a genus (more than 90% of relative abundance according to current scientific evidence), which can lead to colonization by other types of microorganisms and, thus, generate a more diverse environment, which might be related to infertility.

Another important aspect of the present study is the assessment of both diversity and taxonomic characterization according to participants’ history of RIF. Previously, the endometrial and vaginal microbiota were characterized by Kitaya et al. [22] in infertile women with and without history of RIF. The microbiota detected in the endometrium showed significant variation in the composition of the bacterial community between the RIF and non-RIF groups, which was not observed in the vagina. *Burkholderia* spp. were not detected in the microbiota of the endometrium in any sample women without RIF, but they were present in a quarter of the RIF patients. In our study, alpha diversity was higher at the endometrial level in women without RIF. This did not occur in vaginal samples, where no differences were observed. In addition, according to our findings and the available literature, a greater microbial diversity in the case of the vaginal and endometrial microbiota seems to be a scenario most unfavorable for the success of IVF treatments. The results of alpha diversity should be interpreted with caution, especially in patients with RIF. A detailed knowledge of the composition of this microbial pattern is necessary in order to know the % relative abundance of the genera, species, and subspecies present in the samples, and even to examine in depth the presence of pathogens that may have an effect on the vagina and/or endometrium. In the above-mentioned study, the authors found differences in the endometrial microbiota in both groups. When we analyzed the taxonomic characterization, we also observed clear differences in the relative abundances of the different genera and species in the endometrial microbiome between women with and without RIF. Unlike Kitaya et al. [19], we found no differences with respect to the genus *Burkholderia*. The generalized linear model showed that the vaginal microbiome pattern in women without RIF changed between different timepoints of the cycle, while in women with RIF it remained stable. This could be due to a possible adaptation of the microbiome pattern as a result of physiological changes occurring during the cycle. The possible causes and mechanisms involved in these modifications are still unknown and unstudied, but this could be a valuable line of future research.

In view of our study’s findings, before infertility treatment, both the vaginal and endometrial microbiomes can be examined as targets to improve pregnancy rates. A detailed knowledge of the composition of the microbiome pattern, together with the alpha and beta diversity analysis, might be helpful to guide the treatments.

### Strengths and Limitations

Strengths of this study include the use of NGS to sequence the hypervariable regions of the 16S ribosome subunit. Another strength was the use of the Tao Brush to obtain the endometrial samples, as this device has a sheath that covers the specimen and closes before removal from the uterus and passage through the vagina; its design thus prevents any possible contamination by microorganisms from the vagina, and avoids the extraction of samples with patterns that are not directly associated with the endometrial microbiome. Moreover, all of the transferred embryos in the participants were previously analyzed via NGS to confirm chromosome normality and homogeneity, thus avoiding any bias due to embryological factors.

The main limitation was the sample size, which was small due to the pilot study design. Other limitations could be potential confounding stemming from recent sexual relations in some participants, which could have influenced the microbial compositions obtained. Finally, we only extracted endometrial samples at one timepoint in the cycle, in order to avoid compromising the treatment outcome and embryo implantation. This decision limited the amount of information we could obtain on the endometrial pattern, as no samples were taken for the other two timepoints. However, this limitation was partially overcome, as the vaginal samples were taken at three timepoints.

Further studies are needed in order to confirm our findings and to clarify the role of antibiotic and/or probiotic treatment in the normalization of the microbiome pattern and its consequences on clinical outcomes. Future research on the endometrial and vaginal microbiomes, along with their effects on and associations with reproductive health and infertility, should take a comprehensive approach, including an analysis of any interactions with the immune system and between different strains, metabolic and transcriptomic variables, and a study of biofilms—all of which are related to implantation and clinical pregnancy rates.

## 5. Conclusions

There were no differences in the diversity of the vaginal and endometrial microbiomes between women who got pregnant and those who did not, but there were differences in their taxonomic characterization.

The vaginal samples collected showed differences during the secretory phase of the cycle previous to FET only for the Faith index between the women with and without RIF diagnosis, with a higher alpha diversity in women with a history of RIF; the relative abundance of the genera in women with RIF was different from those without RIF, at the taxonomic level.

With respect to the endometrial microbiome pattern, we found statistically significant differences between women with and without RIF.

## Figures and Tables

**Figure 1 jcm-10-04063-f001:**
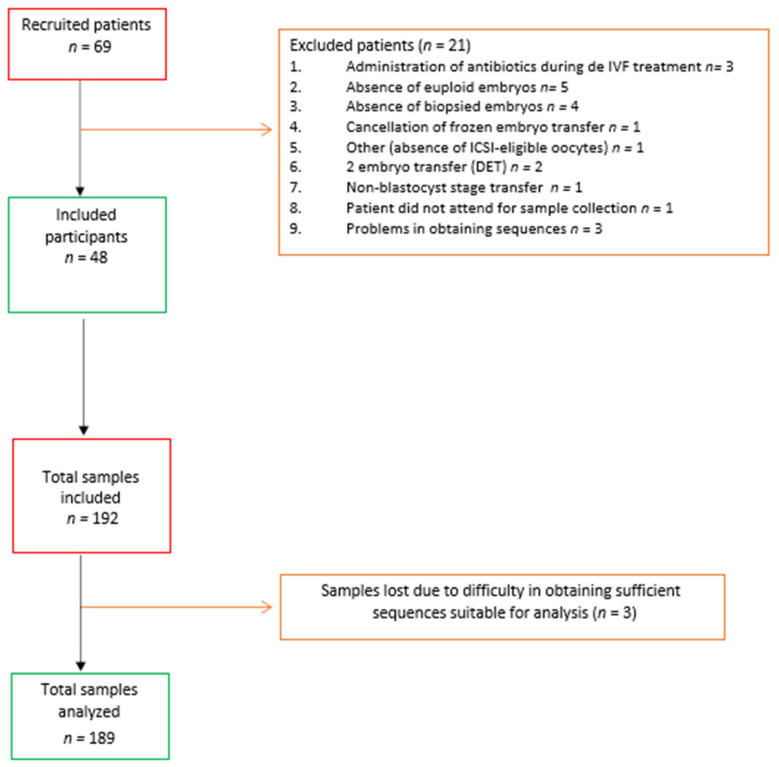
Flow diagram of the study.

**Figure 2 jcm-10-04063-f002:**
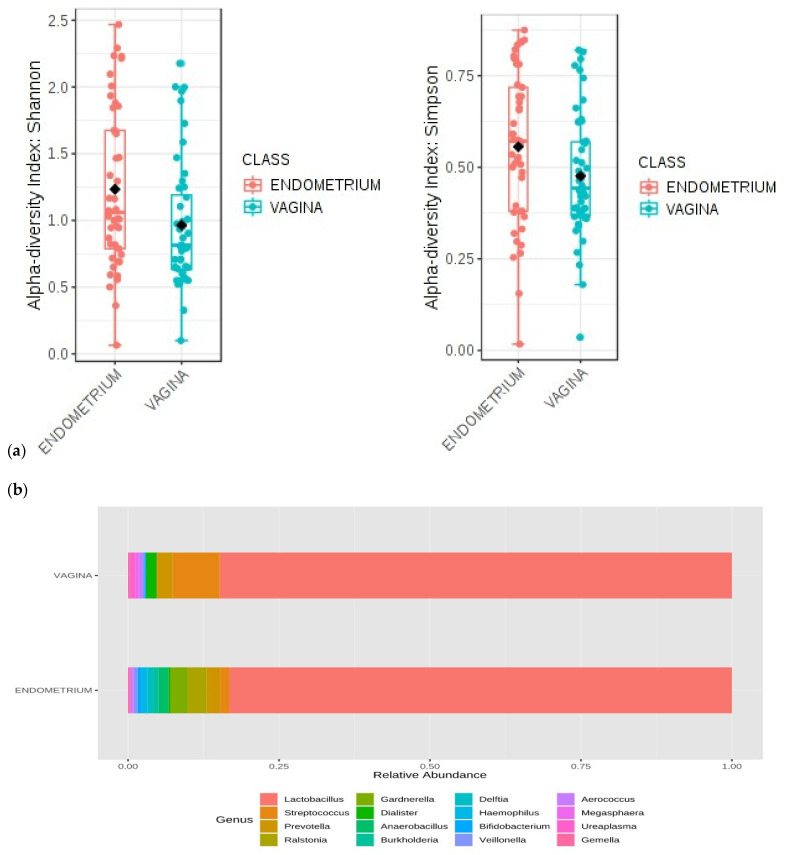
Vaginal and endometrial microbial patterns during the secretory phase of the cycle previous to FET (days 18–22 of the cycle). (**a**) Comparative analysis of the Shannon diversity index and the Simpson diversity index for the women in the study in relation to the study of alpha diversity. (**b**) MicrobiomeAnalyst MDP bar chart of the relative frequencies of the most abundant genera, grouped by type of sample.

**Figure 3 jcm-10-04063-f003:**
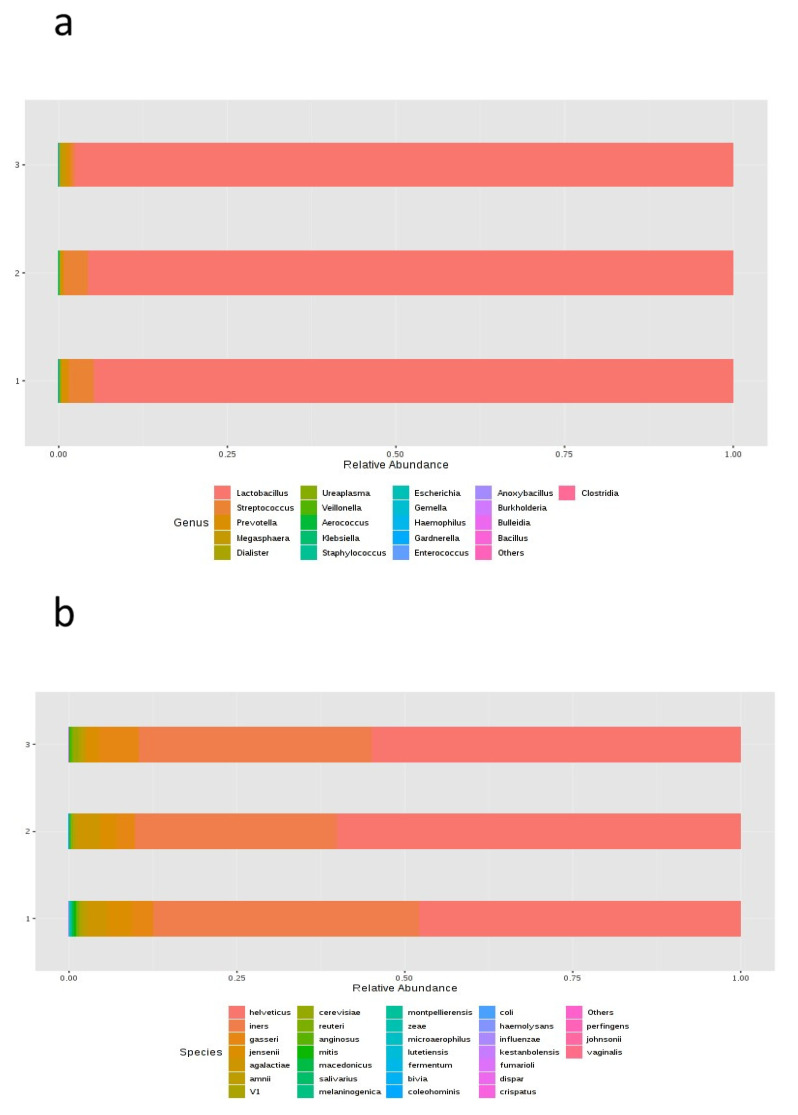
MicrobiomeAnalyst MDP bar chart of the relative frequency of the most abundant (**a**) genera and (**b**) species, grouped by timepoint in the cycle.

**Figure 4 jcm-10-04063-f004:**
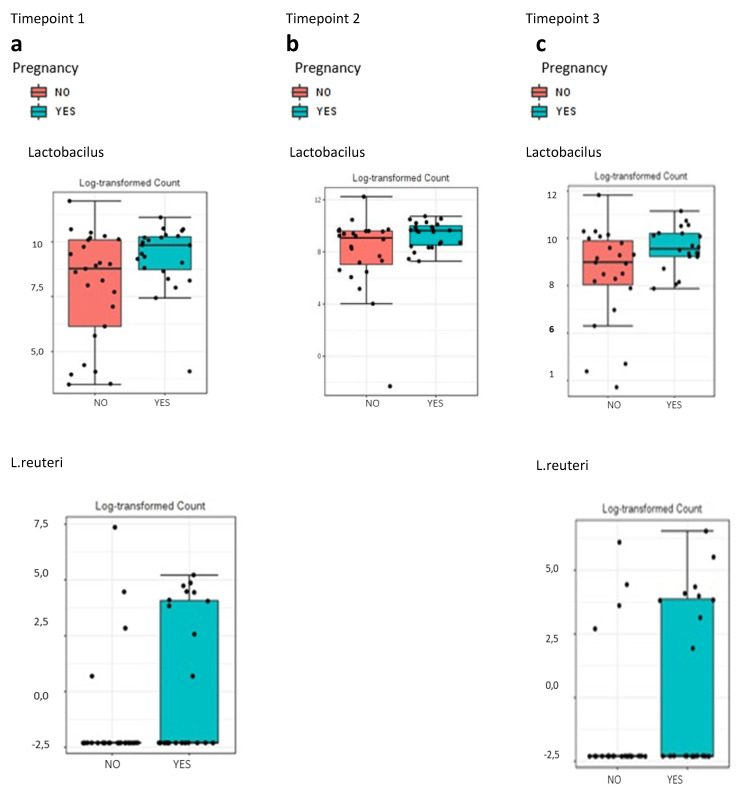
Association of the vaginal samples taken at different timepoints with the gestation rate. (**a**) Univariate analysis represented with box plot showing relative abundance of *Lactobacillus* spp. (0.045) and *L. reuteri* (*p* = 0.040) for the gestation rate during the secretory phase of the cycle previous to frozen embryo transfer (days 18–22 of the cycle). (**b**) Univariate analysis represented with box plot showing relative abundance of the genus *Lactobacillus* spp. (0.027) for the gestation rate on the day of the embryo transfer. (**c**) Univariate analysis represented with box plot showing relative abundance of *Lactobacillus* spp. (*p* = 0.049) and *L. reuteri* (*p* = 0.059) for the gestation rate on the day of the pregnancy test.

**Figure 5 jcm-10-04063-f005:**
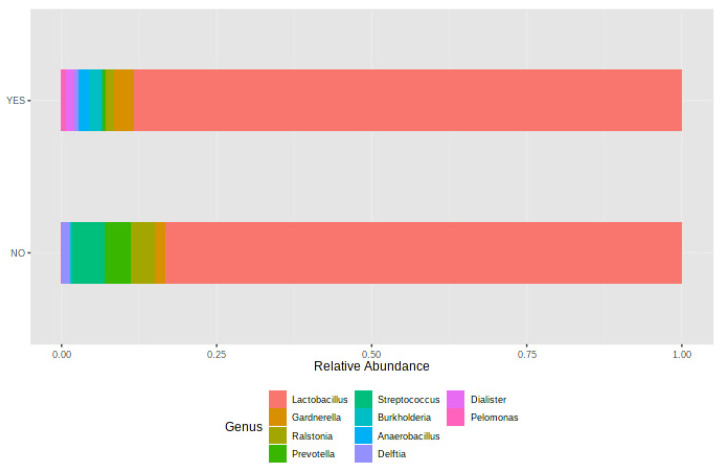
Association of the endometrial sample with the gestation rate: bar chart of the relative frequency of the most abundant genera, grouped by gestation rate.

**Figure 6 jcm-10-04063-f006:**
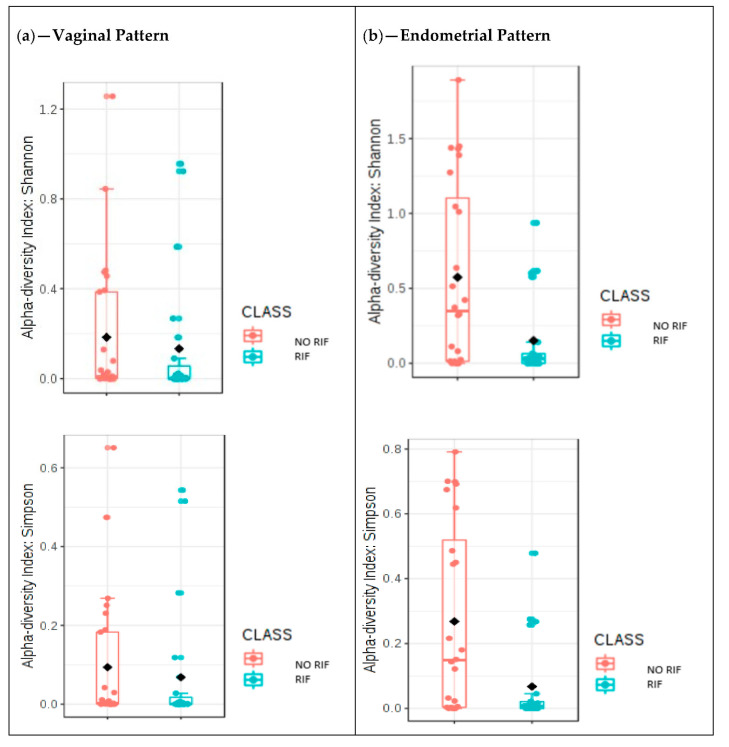
Microbiome patterns by diagnosis of repeated implantation failure (RIF). (**a**) Comparative analysis of the Shannon diversity index (*p* = 0.285) and the Simpson diversity index (*p* = 0.276) for the women with and without RIF in relation to the study of alpha diversity. MicrobiomeAnalyst MDP. (**b**) Univariate analysis repre-sented with box plot showing the differences in the alpha diversity index for the study group in the en-dometrial samples. Shannon diversity index analysis, *p* = 0.021 and Simpson, *p* = 0.021; Mann-Whitney U.

**Figure 7 jcm-10-04063-f007:**
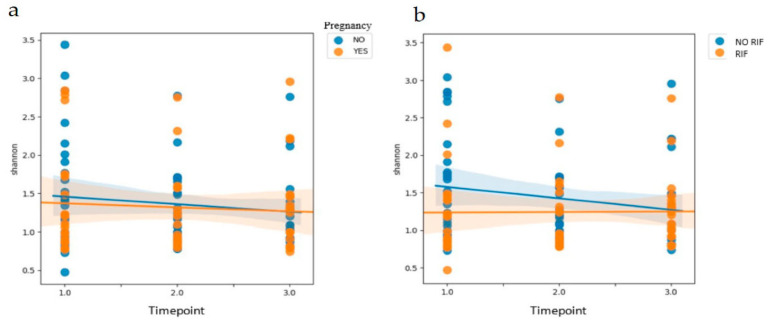
Linear mixed-effects model tests by (**a**) clinical pregnancy outcome and (**b**) the diagnosis of repeated implantation failure (RIF).

**Table 1 jcm-10-04063-t001:** Baseline characteristics in women who achieved and did not achieve clinical pregnancy.

Variables	Total Women *N* = 48	Pregnancy *N* = 21	No Pregnancy *N* = 27	*p*-Value
Age, years, mean (SD)	39.44 (3.82)	38.28 (3.39)	40.13 (3.95)	0.092
Weight, kg, mean (SD)	63.41 (9.79)	59.57 (7.98)	64.69 (10.16)	0.194
Height, cm, mean (SD)	162.33 (6.89)	160.25 (10.41)	163.09 (5.22)	0.481
Tobacco user, *n* (%)	7 (14.29)	4 (16.7)	3 (11.1)	0.598
N of previous pregnancies, mean (SD)	0.67 (0.60)	0.56 (0.51)	0.73 (0.64)	0.296
Previous miscarriages, *n* (%)	28 (58.33)	9 (38.90)	19 (70.0)	0.034
N of previous miscarriages, mean (SD)	1.44 (1.80)	1.11 (1.79)	1.63 (1.81)	0.334
Previous treatments, *n* (%)	37 (77.08)	15 (72.20)	22 (80.0)	0.535
Semen analysis, normozoospermia, *n* (%)	37 (77.08)	17 (83.30)	20 (73.30)	0.617
Donated semen, *n* (%)	10 (20.83)	5 (22.20)	5 (20.0)	0.854
Endometrial thickness, mm, mean (SD)	8.40 (2.04)	8.54 (2.01)	8.32 (2.09)	0.725
Repeated implantation failures, *n* (%)	23 (47.9)	5 (23.81)	18 (66.66)	

SD: standard deviation.

**Table 2 jcm-10-04063-t002:** Differences in the genera present in microbiome profiles in vaginal and endometrial samples.

Genus	Endometrium	Vagina	*p*-Value
*Lactobacillus* spp.	83.17%	84.82%	0.003
*Delftia* spp.	0.95%	0.00%	0.004
*Anaerobacillus* spp.	1.59%	0.00%	0.004
*Ralstonia* spp.	3.17%	0.00%	0.004
*Ureaplasma* spp.	0.00%	0.89%	0.006
*Streptococcus* spp.	1.59%	7.74%	0.019

## Data Availability

All of the data used for this analysis can be confirmed at any time.

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
