# Peer review of "Impact of the Vaginal and Endometrial Microbiome Pattern on Assisted Reproduction Outcomes"

_jcm, 2021, doi:10.3390/jcm10184063_

Round 1

Reviewer 1 Report

The potential impact of the microbiome of the reproductive tract on IVF success has become of increasing interest to clinicians who perform ART procedures. However, very little is known at this time so any contributions to our knowledge of the microbiome of the female reproductive tract is of importance to the field of reproductive medicine. In this paper the authors test both the endometrial and vaginal microbiome through NGS analysis of samples obtained from the endometrial cavity during the secretory phase of a non-treatment cycle prior to an actual FET cycle in which a single PGT embryo is transferred. THe authors also performed vaginal swabs at three timepoints during the cycle.

We really have very little information as to how the vaginal microbiome changes throughout the menstrual cycle. However, it is certainly possible that any changes may reflect confounding influences such as vaginal intercourse or the use of vaginal lubricants during any given menstrual cycle. Does the vaginal microbiome play a role in fertility or impact the success of any fertility treatment or in the pregnancy outcome? Perhaps, but untangling such a complex set of relationships may require a much larger study than this one.

The impact of the endometrium microbiome is perhaps a bit better understood as a result of the two studies that examined the microbiome either at the time of embryo transfer by analysis of the embryo transfer tip or at the time of of sampling for ERA endometrial biopsy at the time of future embryo transfer. Both would seem to me to be more applicable than the timing of endometrial sampling in the present study which does present a weakness in the study design. Does the endometrial biome change in a given patient throughout the menstrual cycle? Maybe. Would it matter compared with what the endometrial biome is at the time of embryo transfer...unlikely.

So the present study adds to our knowledge but I think that the authors need to be careful about overreach in terms of their conclusions concerning the importance of the diversity of the microbiome as opposed to the dominance or non-dominance of lactobaccilus. The analysis of those patients with RIF makes the findings additionally confusing. There is nothing wrong with finding that there are no differences among groups but without any discussion of the power of the study to detect a difference the authors cannot comment on the impact of their findings.

My major concern with the paper as submitted is the extensive figures which are so small as to make them nearly impossible to read. In revising the manuscript I would suggest that the authors reduce the number of figures and enlarge those that are included. For examples I would suggest they consider the figures used in the Moreno (2016) and Franasiak (2016) papers.

Author Response

Dear Reviewer,

First, we would like to thank you for your revision and constructive comments. We believe that the suggestions made are of positive value for our work, contributing to improving the clarity and quality of our paper.

We agree with reviewer that the results of alpha and beta diversity should be interpreted with caution, especially in patients with RIF. This fact does not imply that it is a factor that by itself may be conditioning to suspect the success or failure of the treatment. It is necessary a detailed knowledge of the composition of this microbial pattern, to know the % relative abundance of the genera, species and subspecies present in the samples, and even, to study in depth the presence of pathogens that may have an effect on the vagina and / or endometrium. Therefore, the analysis of alpha and beta diversity, together with that of microbial composition and pathogens, will give us much more information and guide treatment for these patients. We have added at the end of the discussion, before strengths and limitations, this reflection. Please, see line 526.

Regarding the figures, we have reduced the number of figures and we have attached the rest of figures as complementary material. Thank you very much.

The authors.

Reviewer 2 Report

The authors compared the vaginal (at 3 points) and endometrial microbiome patterns between pregnant and non-pregnant women after FET, as well as women with and without RIF. Although the study is retrospective with a relatively sample size, the study is interesting and carefully conducted.

Following concerns came up to my mind.

  1. The legends for Fig. 2,4,7 are missing.
  2. The resolution of Fig. 2d, 3, 4a,c,e, and 6b,c,d,f,g,h is so low that caption in the figures cannot be seen even after magnifying the figures.
  3. L226-230. Difference of alpha and beta diversity between endometrial and vaginal microbiomes. It was an very interesting finding. Please discuss the reason why endometrial microbiome shows higher alpha-diversity, i.e. more richness in bacterial species than vaginal ones, even though the endometrial environment seems more sterile than vagina.
  4. L256-259. No difference in  alpha or beta diversity between the samples over the different time points. Microbiome is usually affected by ovarian steroids and FET cycle may be related to higher local P4 levels than in previous cycle (Time point 1).  Please discuss the effects of ovarian steroids on vaginal/endometrial microbiome.
  5. L277-285.  The authors showed a trend of greater alpha diversity in women who did not get pregnant, with a trend of a negative correlation between the clinical pregnancy rate and alpha diversity. Please discuss these findings, especially in relation to the low endometrial alpha-diversity in RIF patients.
  6. Please discuss what the findings of the present study indicate in a translational point of view. Which should be examined before infertility treatment and should be a target to intervene, vaginal or endometrial microbiome? In case of intervention, would lowering alpha-diversity be helpful to achieve pregnancy? Or would increasing alpha-diversity rescue from RIF? Or should we concern just a specific species without caring about species richness and/or composition of flora?

Author Response

Dear Reviewer,

First, we would like to thank you for your revision and constructive comments. We believe that the suggestions made are of positive value for our work, contributing to improving the clarity and quality of our paper.

Please, find below a point-by-point response to the reviewers’ comments and concerns (their comments in purple font; our response in black font).

  1. The legends for Fig. 2,4,7 are missing.

>> We have added the legends to the new version of the manuscript. Thank you very much.

  1. The resolution of Fig. 2d, 3, 4a,c,e, and 6b,c,d,f,g,h is so low that caption in the figures cannot be seen even after magnifying the figures.

>> We have reduced the number of figures and we have attached the rest of figures as complementary material to improve the quality of the images. Thank you.

  1. L226-230. Difference of alpha and beta diversity between endometrial and vaginal microbiomes. It was an very interesting finding. Please discuss the reason why endometrial microbiome shows higher alpha-diversity, i.e. more richness in bacterial species than vaginal ones, even though the endometrial environment seems more sterile than vagina.

>> Indeed, the diversity of the vaginal microbiome can be expected a priori higher than the endometrial one due to factors, such us the more direct exposure of the vagina to the outside of the body and direct contact with the seminal sample during sexual intercourse. In the present study, we found a higher diversity in the microbiome of the endometrium, and we think that this would be possible due to the non-dominance of a gender (more than 90% of the relative abundance according to current scientific evidence), since this can lead to the colonization of other types of microorganisms and thus an environment with higher diversity is generated, which may be related to infertility factors. This is a hypothesis and it is worth highlighting that the fact that a microbial pattern is more diverse could be a disadvantageous condition with respect to another less diverse. We have tried to clarify this issue in the discussion (line 491). Thank you very much.

  1. L256-259. No difference in alpha or beta diversity between the samples over the different time points. Microbiome is usually affected by ovarian steroids and FET cycle may be related to higher local P4 levels than in previous cycle (Time point 1). Please discuss the effects of ovarian steroids on vaginal/endometrial microbiome.

>> In the present study, it was not possible to take the endometrial sample in the same cycle as the cryotransfe because we must not manipulate the endometrium in the cycle in which the embryo must be implanted in order not to undermine the success of this treatment. Progesterone values ​​can modify the microbial pattern, however, since all patients had one endometrial preparation per hormonal cycle substituted, this factor was homogeneous for all study patients. Thank you for yor comment, we have included it in the discussion section (line 449).

  1. L277-285. The authors showed a trend of greater alpha diversity in women who did not get pregnant, with a trend of a negative correlation between the clinical pregnancy rate and alpha diversity. Please discuss these findings, especially in relation to the low endometrial alpha-diversity in RIF patients.

>> According to our findings and available bibliography, a greater microbial diversity, in the case of the vaginal and endometrial microbiota, seems to be a scenario most unfavorable for the success of IVF treatments. The results of alpha diversity should be interpreted with caution, especially in patients with RIF. This fact does not imply that it is a factor that by itself may be conditioning to suspect the success or failure of the treatment. It is necessary a detailed knowledge of the composition of this microbial pattern, to know the % relative abundance of the genera, species and subspecies present in the samples, and even, to study in depth the presence of pathogens that may have an effect on the vagina and / or endometrium. Therefore, the analysis of alpha and beta diversity, together with that of microbial composition and pathogens, will give us much more information and guide treatment for these patients. We have included this issue in the discussion section (line 510). Thank you.

  1. Please discuss what the findings of the present study indicate in a translational point of view. Which should be examined before infertility treatment and should be a target to intervene, vaginal or endometrial microbiome?

>> It should be chosen based on the patient's medical history. In certain cases,de the endometrial one would be more convenient, when endometrial pathology is already suspected (cases of endometriosis and / or endometritis) and in others only the vaginal one (to optimize the chances of success in implantation, study of dysbiosis in case of failures implantation, or early abortions without endometrial pathology). Even in certain patients it may be necessary and advisable to analyze both the vaginal and the endometrial.

 In case of intervention, would lowering alpha-diversity be helpful to achieve pregnancy? Or would increasing alpha-diversity rescue from RIF? Or should we concern just a specific species without caring about species richness and/or composition of flora?

>> Reducing alpha diversity can occur as a consequence of probiotic treatment in the case of patients with vaginal and / or endometrial dysbiosis, and if the proportion of Lactobacillus spp. increases and becomes more predominant with values ​​greater than 90%, it may have a positive effect on the clinical pregnancy rate. Therefore, the variation in alpha diversity could be a collateral change to the treatment. We do not know its direct relationship with the improvement in rates.

We do not consider that diversity byself may have a direct implication with the RIF patient condition, but this in turn implies having displaced communities of genera such as Lactobacillus spp, and therefore, the growth of other species of microorganisms that otherwise they could not possibly have increased their population (since they would not be in favorable physiological conditions, due to the acidic conditions generated by Lactobacillus spp.).

As we mentioned previously, a detailed knowledge of the composition of the microbiome pattern should be always attached to the alpha and beta diversity analysis.

We have added in the discussion section (line 526) this translational point of view. Thank you very much.

The authors.

Round 2

Reviewer 2 Report

The authors have revised the manuscript appropriately, according to the reviewer's suggestion. Thank you.